# HyperDet: Generalizable Detection of Synthesized Images by Generating and Merging A Mixture of Hyper LoRAs

## Abstract

The emergence of diverse generative vision models has recently enabled the synthesis of visually realistic images, underscoring the critical need for effectively detecting these generated images from real photos. Despite advances in this field, existing detection approaches often struggle to accurately identify synthesized images generated by different generative models. In this work, we introduce a novel and generalizable detection framework termed HyperDet, which innovatively captures and integrates shared knowledge from a collection of functionally distinct and lightweight expert detectors. HyperDet leverages a large pretrained vision model to extract general detection features while simultaneously capturing and enhancing task-specific features. To achieve this, HyperDet first groups SRM filters into five distinct groups to efficiently capture varying levels of pixel artifacts based on their different functionality and complexity. Then, HyperDet utilizes a hypernetwork to generate LoRA model weights with distinct embedding parameters. Finally, we merge the LoRA networks to form an efficient model ensemble. Also, we propose a novel objective function that balances the pixel and semantic artifacts effectively. Extensive experiments on the UnivFD and Fake2M datasets demonstrate the effectiveness of our approach, achieving state-of-the-art performance. Moreover, our work paves a new way to establish generalizable domain-specific fake image detectors based on pretrained large vision models. Our codes are available at `https://anonymous.4open.science/r/HyperDet-3053`.

## 1 Introduction

In recent years, the rapid advancement of generative models—including GANs (Goodfellow et al., 2014), VAEs (Brock et al., 2019; Ho et al., 2020; Karras et al., 2017; 2019; Rombach et al., 2022; Song & Ermon, 2019), GLOW (Kingma & Dhariwal, 2018), and Diffusion models (Sohl-Dickstein et al., 2015)—has enabled the creation of AI-generated images that are often indistinguishable from real ones to the human eyes. This progress allows users to generate realistic images without specialized knowledge, significantly impacting the entertainment industry. However, the proliferation of such images poses serious threats to public opinion and the authenticity of information. Consequently, there is an urgent need for effective methods to monitor and detect synthetic images, ensuring the integrity of information and the fairness of public discourse.

Early detectors primarily focused on images generated by GAN models (Goodfellow et al., 2014), employing spatial (Marra et al., 2019; Rossler et al., 2019; Yu et al., 2019) or frequency (Chandrasegaran et al., 2021; Dong et al., 2022; Frank et al., 2020; Zhang et al., 2019) features to identify synthetic content. However, these methods often struggle with images produced by newer generative models, such as diffusion models. Consequently, there is a growing trend towards developing generalized detectors capable of effectively identifying fake images from a wider range of sources. For instance, Wang et al. (Wang et al., 2020) enhanced the generalization capabilities of detection methods through data augmentation techniques and the use of large datasets. However, excessive training causes the detector model to overfit specific characteristics of the training data.

Some other methods attempt to utilize semantic information for detection. For example, Ojha et al. (Ojha et al., 2023) adopted a pre-trained CLIP (Radford et al., 2021) model to extract high-level

semantic features in synthetic image detection tasks. Jia et al. (Jia et al., 2024) utilized large language models to detect semantic anomalies in synthetic images. However, semantic discrepancies are often minimal between real and generated images with prevalent generative models. In contrast, NPR (Tan et al., 2024), a recent state-of-the-art detector, can capture forgery traces by upsampling operations, leveraging correlations between local pixels. However, NPR mainly relies on low-level pixel artifact features, neglecting the semantic information, leading to high false positive rates.

This work proposes a novel method termed HyperDet, designed to extract generalized artifacts for effective detection. Instead of directly utilizing Spatial Rich Model (SRM) (Fridrich & Kodovsky, 2012) filters, we group these filters to capture varying levels of pixel artifacts based on their functionality and complexity. Corresponding to each SRM filter group, we leverage learnable LoRAs as expert models, specializing in detecting discernible traces in the textural feature space. To facilitate this, we introduce a hypernetwork (Ha et al., 2016) that generates the optimized weights for the LoRAs (ie hyper LoRAs), enabling adaptive selection while learning shared knowledge and expertise among different LoRA experts. Furthermore, we meticulously design a novel objective function that integrates both low-level pixel artifacts and semantic context, effectively mitigating false positives.

Extensive experiments demonstrate that our method exhibits exceptional generalization ability in synthetic image detection tasks. For instance, on the UnivFD dataset (Ojha et al., 2023), our method outperforms the state-of-the-art (SOTA) (Liu et al., 2024) by **8.12%** accuracy and **0.91 mAP**. On the latest Fake2M dataset (Lu et al., 2024), our approach surpasses the SOTA (Tan et al., 2024) by **5.03%** accuracy and **10.02 mAP**. Additionally, we investigate the robustness of our method against various post-processing operations and analyze its generalization ability across different backbone models, highlighting the effectiveness of CLIP in extracting generalized artifacts. We also present the performance of our method across different dataset sizes and discuss the impact of different LoRA ranks and the fine-tuning of various layers on model performance.

Our main contributions can be summarized as follows:

1) We propose a novel and generalizable synthesized image detection method, called HyperDet. Unlike existing detectors, we innovatively introduce hypernetwork into the detection framework that generates optimized weights for specialized LoRA experts, facilitating the extraction of generalized discernible artifacts.

2) We propose an SRM filter grouping strategy to capture varying levels of pixel artifacts based on their functionality and complexity. Besides, we propose a novel objective function to balance the pixel and semantic artifacts effectively.

3) HyperDet achieves state-of-the-art detection performance on multiple datasets, surpassing baseline methods by a large margin. Besides, it shows improved robustness against post-processing operations.

## 2 RELATED WORK

**Synthetic image generation.** In the era of large generative models, synthesized images typically refer to visually realistic images generated from random noise or text prompts. Representative approaches include GANs and their variants (Brock et al., 2019; Choi et al., 2018; Karras et al., 2017; 2019; Park et al., 2019; Zhu et al., 2017), as well as diffusion models (Yang et al., 2024; Nichol et al., 2021; Rombach et al., 2022). GANs transform random noise into images through a generator and optimize the quality via adversarial training, while diffusion models gradually reconstruct images by denoising. Both techniques excel in generating high-quality synthetic images, meanwhile posing significant challenges for image detection.

**Synthetic image detection.** Early visual forgery detection models primarily focused on images generated by Generative Adversarial Networks (GANs). Mo et al. (Mo et al., 2018) trained a binary classification deep neural network to distinguish between real and GAN-generated facial images. Zhang et al. (Zhang et al., 2019) proposed the AutoGAN model, which automatically simulates the GAN sample generation process and observed that the upsampling module of GANs introduces a "checkerboard artifact" in the frequency domain, leading to the extraction of spectral features for classification. Frank et al. (Frank et al., 2020) analyzed the statistical differences between synthetic and real images in the frequency domain.

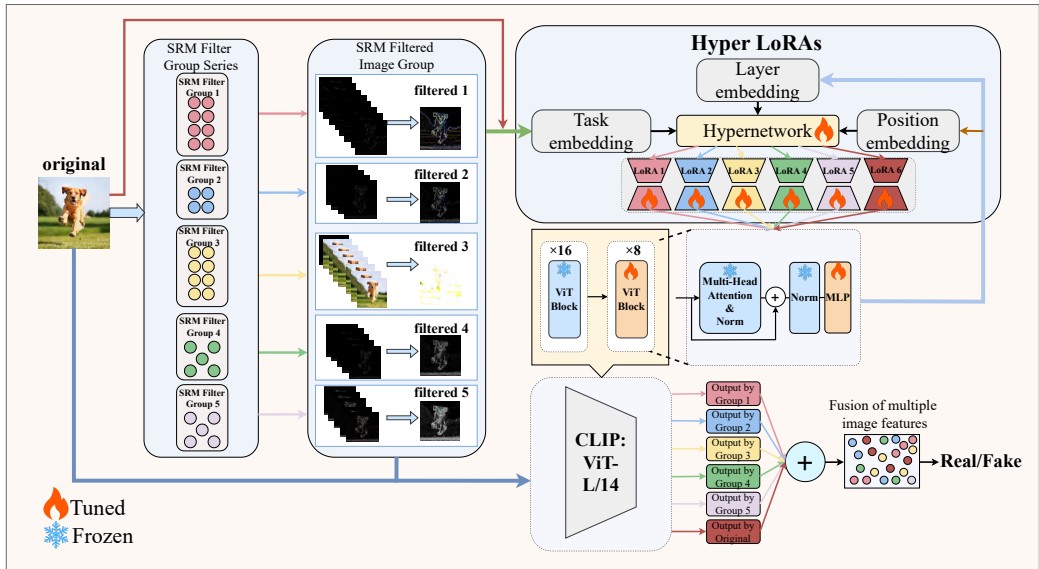

Figure 1: **Overview of the proposed HyperDet framework.** For a given input image, we first generate different filtered views using various groups of filter modules. These filtered views are then used to produce the corresponding task embeddings. Subsequently, the different views, along with the original image, are fed into the ViT module of the CLIP model. Simultaneously, the task embeddings, layer embeddings, and position embeddings are used as inputs to the Hyper LoRAs to generate the corresponding LoRA for fine-tuning CLIP, and finally, the outputs of different LoRA experts are merged to obtain the final output that integrates the knowledge from each expert. This output feature effectively facilitates synthetic image detection.

For facial image forgery detection, Nguyen et al. (Nguyen et al., 2019) utilized capsule networks to identify forgery artifacts. Dang et al. (Dang et al., 2020) proposed an attention-based model to handle important feature maps, enhancing classification capabilities. Liu et al. (Liu et al., 2020) analyzed texture information differences and introduced the Gram-Net model to extract global texture features for detection.

Although these methods perform well on specific datasets, their generalization ability remains limited when confronted with different generative models or some unseen samples. In recent years, researchers have increasingly focused on improving the generalization capability of models. CNNSpot (Wang et al., 2020) detects visual forgeries by identifying the "fingerprints" left by convolutional networks (CNNs) during image generation. The study employs JPEG compression and image blurring as data augmentation techniques, demonstrating that models trained on ProGAN (Karras et al., 2017) synthesized images can generalize effectively to forensic detection across other generative models. Ojha et al. (Ojha et al., 2023) applied $k$-NN and LC classification strategies on a pre-trained CLIP model, achieving good results. Tan et al. (Tan et al., 2024) enhanced low-level artifact detection capabilities by improving the relationships between neighboring pixels. Liu et al. (Liu et al., 2024) employed Moe and LoRA fine-tuning strategies on the CLIP model to enhance generalization performance.

**Low-rank adaption and hypernetwork.** Low-Rank Adaptation (LoRA) (Hu et al., 2021) is an efficient method for fine-tuning large models, particularly pre-trained ones. The core idea is to constrain parameter updates within two low-rank matrices, enabling approximate updates to the original weights. LoRA requires tuning only a small number of additional parameters, thereby preserving pre-trained knowledge while improving computational efficiency and inference speed, allowing the model to better adapt to specific tasks.

In contrast, hypernetworks (Ha et al., 2016) generate model parameters to capture shared knowledge across multiple tasks. Instead of directly fine-tuning the target network, a HyperNetwork learns to generate parameters for different tasks, facilitating cross-task shared learning. This mechanism enhances model flexibility and generalization while reducing training resource consumption.

## 3 METHODOLOGY

In this section, we introduce the specific details of the proposed HyperDet. As illustrated in Fig. 1, HyperDet builds upon a pretrained CLIP model and innovatively introduces SRM filter grouping, Hyper LoRAs tuning, and merging to capture generalized detection traces of synthesized images.

### 3.1 GROUPING SRM FILTERS

The Spatial Rich Model (SRM) (Fridrich & Kodovsky, 2012) is a steganalysis method based on spatial-domain rich models, primarily used for steganalysis in spatially encoded images. It is a dominant approach in traditional steganalysis that relies on handcrafted feature extraction. In synthetic image detection, it can effectively extract pixel artifacts from high-frequency components.

**Grouping strategies for SRM.** Many previous studies(Sun et al., 2022; Zhong et al., 2023) have extensively utilized SRM filters in synthetic image detection. However, our research has found that solely relying on simple filtering, while it can improve detection capabilities to some extent, offers only limited enhancement in performance. In the method proposed in this paper, we designed a novel SRM filter grouping strategy to enhance feature extraction performance. Specifically, the 30 filters are divided into five groups, each characterized by distinct structural features and functions. This classification is based on the functionality and complexity of the filters, dividing them into different groups to better capture various levels of features in the image. The specific classification details can be found in Appendix A. Each group of filters emphasizes different levels of high-frequency texture features.

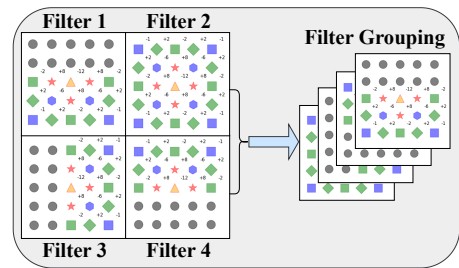

Figure 2: The figure illustrates four filter matrices, each with a size of 5×5. The gray areas indicate matrix elements with a value of zero, while the negative values correspond to the central data to be predicted, and the positive values represent the surrounding data used for prediction. The SRM filter derives residual features by subtracting the central data from the edge data.

Figure 2 illustrates a specific grouping strategy for four of the filters. With the grouped SRM filter sets, given a target image $X$, we apply the filter groups to the image, resulting in residual feature values $R_{ij}$ after filtering. The process can be described as follows:

$$X = \begin{pmatrix} X_{1,1} & \cdots & \cdots & X_{1,m} \\ \vdots & \ddots & X_{i,j} & \vdots \\ X_{n,1} & \cdots & \cdots & X_{n,m} \end{pmatrix} \tag{1}$$

$$R_{ij}^k = \hat{X}_{ij}(\mathcal{N}_{ij}) - X_{ij}, \tag{2}$$

where $R_{ij}^k$ represent the residual value at position $(i, j)$, given that the $k$-th filter in the specific grouping is used, and $\hat{X}_{ij}(\mathcal{N}_{ij})$ represent the estimated value at position $(i, j)$ after filtering based on the neighborhood $\mathcal{N}_{ij}$. $X_{ij}$ is the pixel value at position $(i, j)$ in the original image. Finally, the last residual feature $R_{ij}$ is defined as:

$$R_{ij} = \frac{1}{N} \sum_{k=1}^{N} R_{ij}^k, \tag{3}$$

where $N$ represents the total number of filters in the group, and $R_{ij}^k$ represents the residual feature obtained after the image is processed by the $k$-th filter. As shown in Figure 3, the residual features obtained after processing with the filter bank exhibit more artifacts in the Fourier spectrum compared to the original image. More spectral figures can be found in Appendix B.

### 3.2 HYPER LORAS TUNING

HyperDet leverages Hyper LoRAs to fine-tune the CLIP model for synthetic image detection. As depicted in Figure 1, the network architecture of our proposed method is primarily composed of

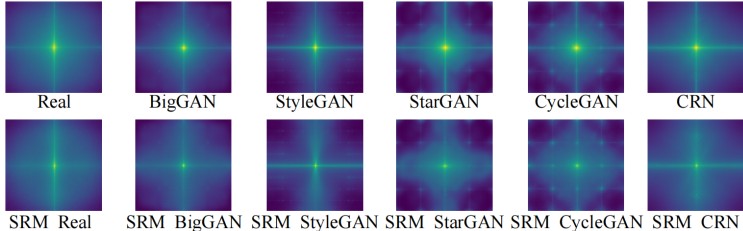

Figure 3: **Frequency analysis of fake and real images.** This figure presents a comparison of the feature maps generated by five models (BigGAN, StyleGAN, StarGAN, CycleGAN, CRN) before and after applying the SRM filter. The top row shows the original feature maps produced by each generative model, while the bottom row displays the corresponding SRM-processed feature maps. After SRM filtering, the edge high-frequency features are enhanced, revealing potential artifacts and inconsistencies, while the central low-frequency features are suppressed, reducing the semantic impact on detection.

three key components: 1) A method for combining SRM filter groups to fuse multiple images, 2) Hyper LoRAs: A hypernetwork that generates corresponding LoRA weights based on three different embeddings and 3) A CLIP model used for fine-tuning. For our method, we first pass the image inputs through five different combinations of SRM filters to extract the corresponding feature embeddings and image input data. Subsequently, we utilize a hypernetwork to generate LoRA weights tailored to different images and tasks. This approach effectively captures the feature differences among images and optimizes the model in a task-adaptive manner, enhancing performance across various tasks. Below, we will provide a detailed explanation of the entire network workflow.

Unlike previous research utilizing hypernetwork(Mahabadi et al., 2021), we opted for LoRA fine-tuning instead of adapter modules. Each image will generate six different perspective images, five of which are new images produced by SRM filter groups, while the other is the original form of the image itself. For each distinct perspective image, a different task embedding $T = \{t_i\}_{i=1}^6$ will be generated, where $i$ represents the embedding value of each perspective. Similarly, we can obtain the layer embeddings from different ViT blocks in the CLIP: ViT-L/14 model as $L = \{l_j\}_{j=17}^{24}$, and the position embeddings from different positions within the MLPs of each ViT block as $P = \{p_k\}_{k=1}^2$. Here, $j$ represents the indices of the different ViT blocks, and $k$ denotes the positions within the MLP layers. Thus, we obtain three distinct embeddings corresponding to the three different parameter inputs required by the hypernetwork. Based on these inputs and the network requirements, we can generate the corresponding LoRA. This approach allows the hypernetwork to learn common knowledge across different tasks, thereby enhancing the model's generalization capability. Specifically, the generation of the LoRA network can be defined as:

$$I = h(T, L, P) \tag{4}$$
$$I_{i,j,k} = h(t_i, L_j, P_k) = (W_A, W_B)_{t_i, L_j, P_k} \tag{5}$$

where $h$ represents a hypernetwork that generates LoRA networks based on three different embedding parameters $(t_i, L_j, P_k)$. $I_{i,j,k}$ represents three distinct embeddings processed through a simple linear network to produce the corresponding network modules.

The principle behind LoRA fine-tuning is that existing large-scale models typically exhibit parameter redundancy, particularly when applied to specific downstream tasks, where only a small subset of parameters plays a major role. Therefore, during the fine-tuning process for specific downstream tasks, the number of parameters to be tuned can be reduced to enhance efficiency. The most commonly used method to achieve this is through low-rank matrix decomposition. Specifically, for a given layer in the network with parameters of size $W \in \mathbb{R}^{d \times d}$, a bypass structure is introduced in which the product of two matrices, $A$ and $B$. Here, the matrix $A$ has a parameter size of $W \in \mathbb{R}^{d \times r}$, and the matrix $B$ has a parameter size of $W \in \mathbb{R}^{r \times d}$, where $r \ll d$. This bypass structure significantly reduces the number of parameters compared to the original model, allowing the original network structure to remain frozen while fine-tuning only the parameters of matrices $A$ and $B$:

$$h = Wx + \Delta W x \tag{6}$$

where $\Delta W$ represents the updated parameters from the bypass network:

$$\Delta W x = W_A W_B x \tag{7}$$

This method effectively leverages the model's parameter compression properties, enhancing the efficiency of fine-tuning.

### 3.3 LOSS FUNCTIONS AND MODEL MERGING

In the synthetic image detection task, we define it as a binary classification problem, where authentic images are labeled as 0 and synthetic images are labeled as 1. This setup allows us to train and evaluate classification models by contrasting the features of authentic and synthetic images to distinguish between them.

We added a sigmoid activation layer to the final output of the model and fine-tuned the last classification layer(Ojha et al., 2023), and using of the cross-entropy loss function for binary classification tasks:

$$\mathcal{L}_{\text{bce}} = -\sum_{f_i \in \mathcal{F}} \log(\psi(\phi_{f_i})) - \sum_{r_i \in \mathcal{R}} \log(1 - \psi(\phi_{r_i})) \tag{8}$$

where the feature vectors have 768 dimensions, with $\phi_{f_i}$ representing the features of synthetic images and $\phi_{r_i}$ denoting the features of real images, and $\psi$ is a classification layer.

The use of a single cross-entropy loss $\mathcal{L}_{\text{bce}}$ may result in unbalanced model optimization, leading to catastrophic forgetting of features from real images. This setup may result in false positives, where authentic images are incorrectly classified as synthetic images. To mitigate this issue, we introduce a total loss that incorporates both the original images and the filtered images. $\mathcal{L}$ is computed as:

$$\mathcal{L} = \alpha \mathcal{L}_{\text{bce}}^{O} + (1 - \alpha) \mathcal{L}_{\text{bce}}^{F} \tag{9}$$

$\mathcal{L}_{\text{bce}}^{O}$ represents the binary cross-entropy loss derived from the original image without applying any filters, while $\mathcal{L}_{\text{bce}}^{F}$ denotes the binary cross-entropy loss obtained after the image is processed by one of the five groups of filter combinations. The parameter $\alpha$ is a hyperparameter, typically set to a small value, and is set to 0.1 in this study. This is because the model needs to focus on processing the rich texture features obtained after filtering, while also ensuring the preservation of features from the original image, thereby maintaining the network's ability to detect real images. The specific feature fusion algorithms used in training and inference are detailed in Appendix F.

## 4 EXPERIMETNS

### 4.1 SETTINGS

**Implementation details.** The study employs Hyper LoRAs modules to fine-tune the last 8 fully connected layers of the CLIP: ViT-L/14 model, with the generated LoRA's rank set to 16. To ensure experimental consistency (Ojha et al., 2023; Wang et al., 2020; Liu et al., 2024), a dataset comprising 720k images across 20 categories was used (including 360k real images and 360k synthetic images generated by ProGAN (Karras et al., 2017)). During training, the probabilities for Gaussian blur and JPEG compression data augmentation were set to 0.1, and the learning rate was fixed at 0.0001. The model was trained for 5 epochs. Experiments were conducted on a server with two RTX 4090 GPUs.

**Datasets.** We evaluate the generalization capability of our approach on the UnivFD dataset (Ojha et al., 2023), which contains 19 different types of data, and the Fake2M dataset (Lu et al., 2024), which includes 17 different types of data. The UnivFD dataset contains the generators from: ProGAN (Karras et al., 2017), CycleGAN (Zhu et al., 2017), BigGAN (Brock et al., 2019), StyleGAN (Karras et al., 2019), GauGAN (Park et al., 2019), StarGAN (Choi et al., 2018), Deepfakes (Rossler et al., 2019), SITD (Chen et al., 2018), SAN (Dai et al., 2019), CRN (Chen & Koltun, 2017), IMLE (Li et al., 2019), Guideed (Dhariwal & Nichol, 2021), LDM (Rombach et al., 2022), Glide (Nichol et al., 2021), DALL-E (Ramesh et al., 2021). Additionally, we conducted robustness evaluations on these datasets and performed various ablation experiments. Fake2M is a recently collected, larger-scale synthetic image dataset, primarily consisting of data generated by the following generators: Stable Diffusion model (Rombach et al., 2022), Midjourney (Midjourney), Cogview (Ding et al., 2021), StyleGan (Karras et al., 2019). This dataset contains synthetic images generated by various diffusion models, which exhibit more realistic visual effects. Previous methods have shown detection performance on this dataset that is nearly at the level of random chance.

**Evaluation protocol.** We follow the evaluation protocol of previous work (Chai et al., 2020; Ojha et al., 2023; Wang et al., 2020; 2023), by adopting mean Average Precision (mAP) and classification accuracy (avg. Acc) as the primary metrics for assessing our detection method. For all datasets, robustness tests, and ablation studies, we report both mean Average Precision (mAP) and average accuracy (avg. Acc) as evaluation metrics. We have visualized our superior discrimination performance using t-SNE in Appendix C. To explore the advantages of Hyper LoRAs in knowledge sharing and model merging, we compared it with the MoE of LoRA and standalone models in Appendix D. Our baselines include:**CNNSpot** (Wang et al., 2020) fine-tunes a ResNet-50 (He et al., 2016), which was pre-trained on ImageNet (Deng et al., 2009), **PatchForensics** (Chai et al., 2020) uses Xception (Chollet, 2017) to train a fully-convolutional patch-level classifier with restricted receptive fields, **CoOccurrence** (Nataraj et al., 2019) utilizes co-occurrence matrices to train a classifier for distinguishing between real and fake images, **FreqSpec** (Zhang et al., 2019) identifies sampling artifacts in the frequency spectra of GAN-generated images, **DIRE** (Wang et al., 2023) detects diffusion-generated images by leveraging the reconstruction error from the pre-trained ADM (Dhariwal & Nichol, 2021), **UnivFD** (Ojha et al., 2023) employs the pre-trained features from the CLIP (Radford et al., 2021) visual encoder for classification through nearest neighbor and linear probing techniques, **NPR** (Tan et al., 2024) identifies that these operations create strong correlations among local pixels during image generation. By capturing the artifacts introduced by upsampling operations through Neighboring Pixel Relationships (NPR), these techniques effectively detect such pseudo-artefacts, **MoE for ViT-L/14** (Liu et al., 2024), with the application of Mixture of Experts (MoE) and Low-Rank Adapters (LoRA), the Vision Transformer (ViT) layers within the CLIP model were fine-tuned to enhance the performance of synthetic image detection.

Table 1: **Generalization accuracy results on UnivFD dataset (Ojha et al., 2023).** The classification accuracy (acc) results for different methods of detecting fake images indicate that models outside the GAN column can be considered as belonging to the generalization domain. Our method, HyperDet, demonstrates a significant improvement in generalization performance, achieving an overall increase of +8.12 acc compared to other methods.

| Method | Generative adversarial networks | | | | | | Deep-fakes | Low level vision | | Perceptual loss | | Diffusion models | | | | | | | | Total |
| | Pro-GAN | Cycle-GAN | Big-GAN | Style-GAN | Gau-GAN | Star-GAN | | SITD | SAN | CRN | IMLE | Guided | LDM 200s | LDM 200s w/CFG | LDM 100s | GLIDE 100-27 | GLIDE 50-27 | GLIDE 100-10 | DALL-E | avg. Acc |
|---|---|---|---|---|---|---|---|---|---|---|---|---|---|---|---|---|---|---|---|---|
| CNNSpot (Wang et al., 2020) | **100.0** | 80.49 | 55.77 | 64.14 | 82.23 | 80.97 | 50.66 | 56.11 | 50.00 | 87.73 | 92.85 | 52.30 | 51.20 | 52.20 | 51.40 | 53.45 | 55.35 | 54.30 | 52.60 | 64.41 |
| PatchForensics Chai et al. (2020) | 68.81 | 53.02 | 55.76 | 59.24 | 52.64 | 77.49 | 55.78 | 59.65 | 48.80 | 65.57 | 61.69 | 52.26 | 58.53 | 60.72 | 58.21 | 55.78 | 56.58 | 55.05 | 61.24 | 58.78 |
| CoOccurrence Nataraj et al. (2019) | 97.70 | 63.15 | 53.75 | 92.50 | 51.10 | 54.70 | 57.10 | 63.06 | 55.85 | 65.65 | 65.80 | 60.50 | 70.70 | 70.55 | 71.00 | 70.25 | 69.60 | 69.90 | 67.55 | 66.86 |
| FreqSpec Zhang et al. (2019) | 49.90 | **99.90** | 50.50 | 49.90 | 50.30 | 99.70 | 50.10 | 50.00 | 48.00 | 50.60 | 50.10 | 50.90 | 50.40 | 50.40 | 50.30 | 51.70 | 51.40 | 50.40 | 50.00 | 55.50 |
| DIRE Wang et al. (2023) | **100.0** | 67.73 | 64.78 | 83.08 | 65.30 | **100.0** | **94.75** | 57.62 | 60.96 | 62.36 | 62.31 | **83.20** | 82.70 | 84.05 | 84.25 | 87.10 | **90.80** | **90.25** | 58.75 | 77.89 |
| UnivFD Ojha et al. (2023) | **100.0** | 98.25 | 95.00 | 84.75 | 99.40 | 95.50 | 69.55 | 64.90 | 56.50 | 57.00 | 67.90 | 69.70 | 93.25 | 72.75 | 93.90 | 77.30 | 77.85 | 76.80 | 86.15 | 80.82 |
| NPR Tan et al. (2024) | 99.90 | 95.20 | 84.00 | **98.85** | 80.90 | 99.80 | 77.20 | 55.60 | 64.40 | 50.00 | 50.00 | 74.00 | 80.60 | 80.40 | 80.70 | 79.80 | 79.90 | 80.20 | 73.40 | 78.15 |
| MoE for ViT-L/14 Liu et al. (2024) | **100.0** | 99.58 | 96.10 | 90.10 | **99.70** | 99.05 | 54.95 | 84.00 | 55.50 | 76.95 | 91.30 | 65.15 | 94.65 | 74.30 | 95.60 | 78.75 | 81.40 | 80.55 | 86.05 | 83.98 |
| **HyperDet (ours)** | **100.0** | 97.40 | **97.50** | 97.50 | 96.20 | 98.65 | 73.85 | **93.00** | 75.00 | 92.75 | 93.20 | 77.35 | 98.70 | 96.60 | 98.80 | 87.75 | 89.95 | 88.70 | 97.00 | 92.10 |

Table 2: **Generalization average precision results on UnivFD dataset (Ojha et al., 2023).** The average precision (AP) results for different methods of detecting fake images indicate that models outside the GAN column can be considered as belonging to the generalization domain. Our method, HyperDet, demonstrates a significant improvement in generalization performance, achieving an overall increase of +0.91 mAP compared to other methods.

| Method | Generative adversarial networks | | | | | | Deep-fakes | Low level vision | | Perceptual loss | | Diffusion models | | | | | | | | Total |
| | Pro-GAN | Cycle-GAN | Big-GAN | Style-GAN | Gau-GAN | Star-GAN | | SITD | SAN | CRN | IMLE | Guided | LDM 200s | LDM 200s w/CFG | LDM 100s | GLIDE 100-27 | GLIDE 50-27 | GLIDE 100-10 | DALL-E | mAP |
|---|---|---|---|---|---|---|---|---|---|---|---|---|---|---|---|---|---|---|---|---|
| CNNSpot (Wang et al., 2020) | **100.0** | 96.36 | 85.34 | 98.10 | 98.48 | 96.97 | 60.33 | 82.95 | 54.22 | **99.61** | 99.81 | 69.93 | 66.17 | 67.68 | 66.13 | 71.18 | 76.37 | 72.13 | 67.66 | 80.50 |
| PatchForensics Chai et al. (2020) | 68.44 | 55.59 | 64.37 | 64.10 | 58.74 | 84.48 | 59.92 | 72.08 | 47.63 | 73.05 | 76.87 | 58.98 | 77.05 | 76.87 | 76.35 | 75.97 | 77.41 | 74.68 | 71.91 | 68.74 |
| CoOccurrence Nataraj et al. (2019) | 99.74 | 80.95 | 50.61 | 98.63 | 53.11 | 67.99 | 59.14 | 68.98 | 60.42 | 73.06 | 87.21 | 70.20 | 91.21 | 89.02 | 92.39 | 89.32 | 88.35 | 82.79 | 80.96 | 78.11 |
| FreqSpec Zhang et al. (2019) | 55.39 | **100.0** | 75.08 | 55.11 | 66.08 | **100.0** | 45.18 | 47.46 | 57.12 | 53.61 | 50.98 | 57.72 | 77.72 | 77.25 | 76.47 | 68.58 | 64.58 | 61.92 | 67.77 | 66.21 |
| DIRE Wang et al. (2023) | **100.0** | 76.73 | 72.80 | 97.06 | 68.44 | **100.0** | **98.55** | 54.51 | 65.62 | 97.10 | 93.74 | 94.29 | 95.17 | 95.43 | 95.77 | 96.18 | 97.30 | 97.53 | 68.73 | 87.63 |
| UnivFD Ojha et al. (2023) | **100.0** | 99.76 | 99.31 | 97.48 | 99.98 | 99.28 | 83.12 | 64.10 | 76.38 | 97.64 | 98.40 | 87.64 | 98.68 | 89.65 | 98.7 | 92.84 | 93.22 | 92.33 | 96.07 | 92.80 |
| NPR Tan et al. (2024) | **100.0** | 95.10 | 85.60 | **99.90** | 83.00 | **100.0** | 76.00 | 60.50 | 66.00 | 50.00 | 50.00 | 78.10 | 85.40 | 85.40 | 85.30 | 85.40 | 85.70 | 86.00 | 76.30 | 80.72 |
| MoE for ViT-L/14 Liu et al. (2024) | **100.0** | 99.85 | 99.88 | 99.69 | **100.0** | 99.68 | 87.38 | 88.26 | 84.48 | 98.82 | 99.84 | 93.39 | 99.81 | 96.80 | 99.88 | 98.71 | 98.84 | 98.60 | 98.81 | 96.99 |
| **HyperDet (ours)** | **100.0** | 99.96 | 99.89 | 99.73 | 99.93 | **100.0** | 88.38 | 97.12 | 89.22 | 98.82 | 99.98 | 95.31 | 99.86 | 99.14 | 99.90 | 97.20 | 97.99 | 98.02 | 99.65 | 97.90 |

## 4.2 GENERALIZATION ACROSS SYNTHETIC IMAGE DATASETS

**Evaluation on the UnivFD dataset.** We conducted a comparative evaluation of our method against several state-of-the-art synthetic image detectors.

Tables 1 and 2 present the accuracy and average precision results of different detectors (rows) across various generators (columns). The values reported represent the average performance for each model. It can be observed from the results that while some models have exhibited a certain degree of generalization capability, their performance remains limited, particularly showing a significant decline in certain datasets. For instance, in more complex diffusion models such as GLIDE, traditional detection methods like UnivFD (Ojha et al., 2023) and CNNSpot (Wang et al., 2020) often experience a sharp decline in performance. This is primarily due to these detectors' inability to effectively capture the inherent low-level texture features. Additionally, some network architectures, such as NPR (Tan et al., 2024), overly emphasize the role of low-level artifact, resulting in a high number of false positives and limited generalization performance. To address this issue, our method (HyperDet) utilizes a combination of five groups of SRM filters to extract low-level artifact features while retaining the original image features during model fusion. This approach ensures improved generalization capability for detecting synthetic images while effectively mitigating false positives.

**Evaluation on the Fake2M dataset.** The Fake2M dataset contains images generated by the latest state-of-the-art generators, whose realism has reached a level where they are nearly indistinguishable from the naked eye and perform poorly on many traditional detectors. We evaluated the aforementioned baseline models on this dataset.

Figures 4a and 4b demonstrate that other methods achieve lower mean average precision (mAP) and average accuracy (avg.ACC) on the novel Fake2M dataset, underscoring the challenges that new datasets present to generalizable detection methods. In contrast, our method surpasses all baseline approaches across both metrics, delivering a notable improvement of +5.03% acc and +10.02mAP.

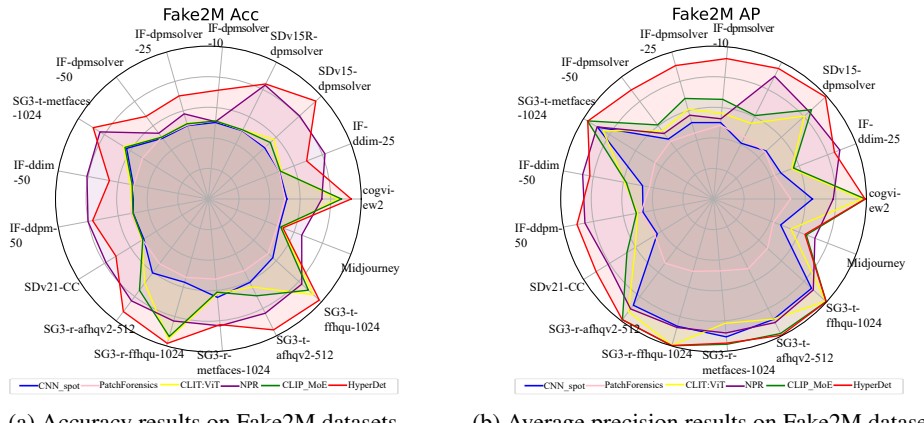

(a) Accuracy results on Fake2M datasets      (b) Average precision results on Fake2M datasets

Figure 4: **Generalization results on Fake2M dataset (Lu et al., 2024).** We used radar charts to present the detection of accuracy results, with each concentric circle representing a 20% scale. Our method demonstrated optimal performance across multiple datasets. In Midjourney, the performance exhibits slightly inferior results.

### 4.3 ROBUSTNESS AGAINST POST-PROCESSING OPERATIONS

Images often undergo various post-processing operations during transmission, which can impact detection performance. To demonstrate the robustness of our method, we evaluated the performance of our method on the UnivFD dataset, specifically focusing on two common post-processing techniques: Gaussian blur (sigma = 1, 2, 3, 4) and JPEG compression (quality = 90, 80, 70, 60, 50, 40, 30). We compared the robustness of our proposed method against CNNspot (Wang et al., 2020), UnivFD (Ojha et al., 2023), and NPR (Tan et al., 2024).

Figures 5 and 6 present the robustness evaluation results of the four models. We found that our method achieved excellent performance against Gaussian blur, primarily due to the application of filters that extract a wide range of low-level artifact features. In the case of JPEG compression, our method also performed well, although some performance metrics were slightly less comparable to those of the UnivFD method.

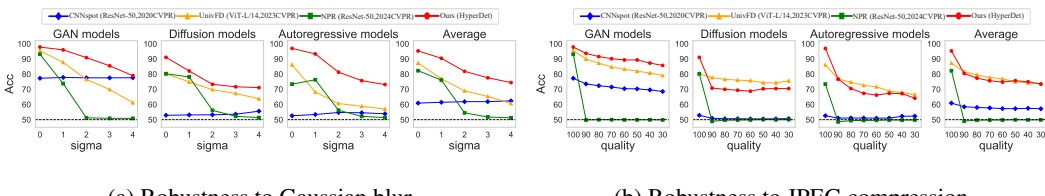

(a) Robustness to Gaussian blur

(b) Robustness to JPEG compression

Figure 5: **Robustness evaluation results of accuracy.** We conducted robustness evaluations on four detection methods under two post-processing conditions: (a) Gaussian blur and (b) JPEG compression, using the UnivFD dataset. The results indicate that our method (HyperDet) outperforms other networks across all post-processing scenarios involving Gaussian blur, while underperforming slightly compared to the UnivFD method in some cases of JPEG compression.

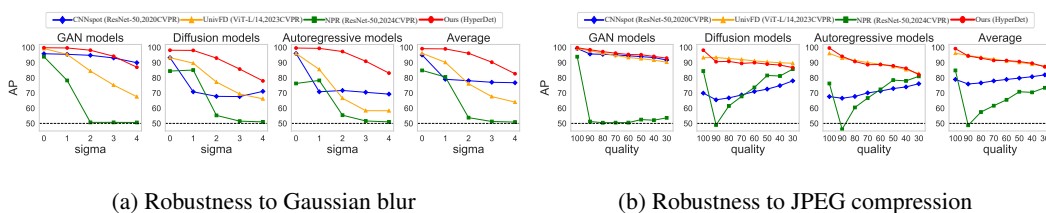

(a) Robustness to Gaussian blur

(b) Robustness to JPEG compression

Figure 6: **Robustness evaluation results of average precision.** As shown in Figure 5, the average precision metric generally exhibits similar performance across the methods.

### 4.4 IMPACTS OF DIFFERENT BACKBONE NETWORKS

Our goal is to leverage pre-trained visual networks to enhance the generalization capability of synthetic image detection. Therefore, we compared the performance of various pre-training settings across different ViT architectures. In addition to various ViT variants of CLIP, we also considered a range of network architectures pre-trained on ImageNet-21k (Deng et al., 2009), keeping all other settings consistent. As shown in Figure 7, the CLIP visual encoders exhibit significantly better detection performance. Moreover, larger models generally demonstrate stronger capabilities. Notably, CLIP: ViT-L/14, with its excellent pre-training results and extensive network structure, allows for broader generalization to diverse datasets, leading to a better understanding of the distribution of natural data and a deeper learning of the underlying details that distinguish real from synthetic images.

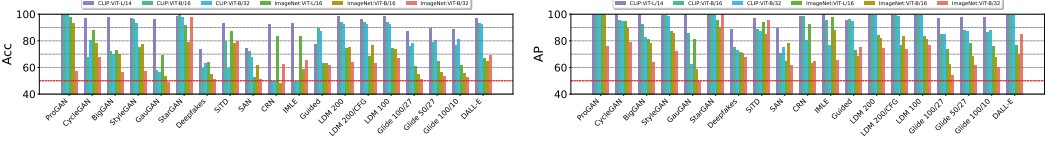

(a) Accuracy of different pre-trained backbones          (b) Average precision of different pre-trained backbones

Figure 7: **The impact of different backbone networks.** Compared to ImageNet pre-trained models, CLIP pre-trained models significantly enhance the generalization capability across various variants, exhibiting higher accuracy and average precision. The red dashed line in the figure represents the random performance baseline.

### 4.5 EFFECTS OF TRAINING DATA SCALE

Our experiments utilized ProGAN as the training data, consisting of 360k synthetic images and 360k real images, totaling 720k images. We observed that with this data size, HyperDet achieved good generalization in detection performance. In this section, we investigate whether HyperDet can still maintain strong generalization capabilities with smaller datasets. To this end, we evaluated HyperDet on datasets with sizes of 2k, 8k, 20k, 80k, 200k, and 720k. Figure 8 illustrates the generalization performance of HyperDet across various data scales. We found that HyperDet maintains good

generalization even on smaller datasets. However, when the data size is extremely small (e.g., 2k), the performance declines significantly. This is primarily due to overfitting on the specific dataset, which leads to a substantial bias and diminishes the advantages of CLIP features.

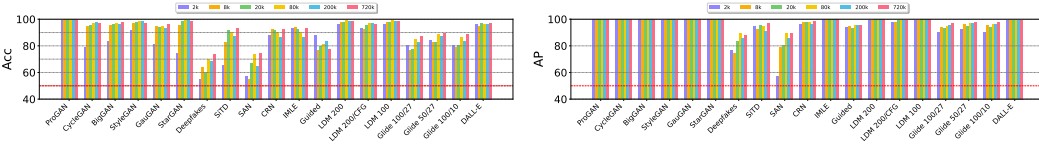

(a) Accuracy of different datasets scale      (b) Average precision of different datasets scale

Figure 8: **Effect of training datasets scale.** We evaluated the performance of the CLIP: ViT-L/14 variant in synthetic image detection across different data scales. The results show a positive correlation between overall performance and data scale. However, our method still maintains a certain level of generalization even on smaller datasets.

### 4.6 ABLATION STUDY ON DIFFERENT LAYERS AND LoRA RANKS

We found that fine-tuning the 8 MLP layers in ViT-L/14 has already achieved good results. This raises the question: what would the outcome be if we adjusted more or fewer layers? Additionally, in previous experiments, we set the LoRA rank to 16; how would changing the rank to other values impact the network's performance? This section presents the performance of our method (HyperDet) under different configurations. Table 3 shows the superior accuracy performances under different network depths and varying ranks of LoRA, and mAP comparisons can be found in Appendix E. The experimental results demonstrate the robustness and generalization of our method.

Table 3: **Accuracy under different numbers of layers and ranks.** The left two columns represent the combinations of different numbers of layers and ranks. Throughout the experiment, nearly every combination exhibited effective generalization performance.

| Fine-tuned layers | LoRA rank | Pro-GAN | Cycle-GAN | Big-GAN | Style-GAN | Gau-GAN | Star-GAN | Deep-fakes | SITD | SAN | CRN | IMLE | Guided | LDM 200s | LDM 200s w/CFG | LDM 100s | GLIDE 100-27 | GLIDE 50-27 | GLIDE 100-10 | DALL-E | avg. Acc |
|---|---|---|---|---|---|---|---|---|---|---|---|---|---|---|---|---|---|---|---|---|---|
| | 4 | **100.0** | 99.00 | 98.55 | 93.20 | 96.05 | **99.85** | 59.30 | **96.00** | 73.00 | 89.85 | 90.15 | 73.60 | 98.20 | 94.30 | 98.50 | 85.10 | 89.35 | 86.30 | 95.05 | 90.28 |
| 7 | 8 | **100.0** | 98.25 | **98.65** | 97.90 | **97.90** | 97.25 | 69.20 | 94.00 | 76.50 | 95.45 | 96.25 | 76.05 | 99.10 | 95.30 | 99.15 | 83.35 | 87.95 | 84.05 | 97.35 | 91.77 |
| | 16 | **100.0** | **99.10** | 98.00 | 97.45 | 97.20 | 99.15 | 58.70 | 92.50 | 75.00 | **97.25** | **97.60** | 82.40 | 99.15 | 96.00 | **99.30** | 84.95 | 89.15 | 86.80 | 96.70 | 91.92 |
| | 32 | **100.0** | 98.30 | 96.95 | 97.50 | 94.60 | 96.65 | 64.35 | 94.50 | 78.50 | 87.60 | 87.75 | 80.50 | 98.75 | 96.40 | 98.80 | 89.15 | 92.20 | 90.95 | 97.30 | 91.62 |
| | 4 | **100.0** | 98.65 | 96.90 | 98.40 | 93.45 | 99.80 | 63.40 | 93.50 | 68.00 | 91.60 | 93.50 | 82.00 | 98.40 | 96.30 | 98.40 | 79.85 | 84.60 | 82.60 | 97.85 | 90.38 |
| 8 | 8 | **100.0** | 99.00 | 96.55 | **98.55** | 92.25 | 99.80 | 74.30 | 92.50 | 73.00 | 90.75 | 91.10 | 82.30 | 97.45 | 95.05 | 97.50 | 84.30 | 87.90 | 86.85 | 96.55 | 91.35 |
| | 16 | **100.0** | 97.40 | 97.50 | 97.50 | 96.20 | 98.65 | 73.85 | 93.00 | 75.00 | 92.75 | 93.20 | 77.35 | 98.70 | 96.60 | 98.80 | 87.75 | 89.95 | 88.70 | 97.00 | **92.10** |
| | 32 | **100.0** | 98.30 | 97.20 | 98.00 | 92.00 | 98.65 | **74.95** | 95.00 | 65.50 | 89.90 | 89.95 | 81.25 | 96.00 | 94.50 | 96.15 | 81.45 | 85.75 | 82.95 | 94.75 | 90.12 |
| | 4 | **100.0** | 98.75 | 97.05 | 95.80 | 94.95 | 99.40 | 59.50 | 94.50 | 72.00 | 90.50 | 91.25 | 78.95 | 98.55 | 97.10 | 99.15 | 76.50 | 82.55 | 77.90 | 97.90 | 89.59 |
| 9 | 8 | **100.0** | 97.45 | 95.25 | 96.80 | 92.30 | 99.55 | 61.35 | 89.50 | 72.50 | 87.85 | 88.35 | 81.45 | 98.60 | 97.10 | 98.95 | 82.40 | 87.55 | 85.50 | 97.55 | 90.00 |
| | 16 | **100.0** | 96.85 | 92.75 | 98.10 | 88.45 | 97.65 | 70.10 | 88.50 | **84.00** | 61.10 | 61.10 | **88.45** | 99.10 | **98.55** | 99.10 | **91.65** | **95.20** | **93.65** | **98.50** | 89.62 |
| | 32 | **100.0** | 99.05 | 95.80 | 98.50 | 92.40 | 99.45 | 63.60 | 91.00 | 78.00 | 73.45 | 73.45 | 82.45 | **99.20** | 98.00 | 99.25 | 85.15 | 89.85 | 87.40 | 98.45 | 89.71 |

## 5 CONCLUSION

In this work, we have developed a generalizable fake imagery detection method termed HyperDet that is particularly effective in distinguishing synthetic images generated by unseen source models. HyperDet first groups SRM filters to enable efficient extraction of pixel artifacts from high-frequency in synthetic images, it then utilizes expert models based on learnable LoRAs to capture corresponding discernible features. Importantly, we introduce Hyper LoRAs, which leverage a hypernetwork to generate weights for different LoRA experts to extract shared knowledge during model learning. Finally, the experts are merged to increase model generalization capabilities. HyperDet effectively reduces false positives and exhibits strong generalization across multiple datasets, contributing to future research in synthetic image detection.

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

## A    THE DETAILS OF THE FILTER GROUPING

The filters are grouped based on their functionality and complexity, with each group designed to capture different levels of image features:

- **Group 1 (Filters 1-8):** These filters focus on simple edge detection, primarily capturing subtle directional changes in the image, including horizontal, vertical, and diagonal edges.
- **Group 2 (Filters 9-12):** Filters in this group introduce stronger weight variations to emphasize more complex edge features, especially those that are more prominent and defined.
- **Group 3 (Filters 13-20):** This group is designed to recognize multi-level edges and curved structures. The filters are more complex and focus on extracting high-frequency texture information with significant directional changes.
- **Group 4 (Filters 21-25):** Filters in this group aim to extract large-scale features, identifying coarse edges and contours that highlight significant structures, particularly in the low-frequency range.
- **Group 5 (Filters 26-30):** This group contains high-order edge detection and texture extraction filters. These filters are well-suited for capturing fine textures and large-scale directional features, useful in detailed texture analysis.

By grouping the filters in this manner, the model can effectively analyze diverse image characteristics across multiple scales, enhancing the overall performance of feature extraction.

## B    MORE AVERAGED SPECTROGRAM PRESENTATIONS.

In Figure 9, we present additional spectrograms processed by the filter groups, along with the corresponding unfiltered spectrograms. A comparative analysis indicates that the filtering process results in a significant attenuation of signal characteristics in the low-frequency regions while enhancing the features in the high-frequency regions.

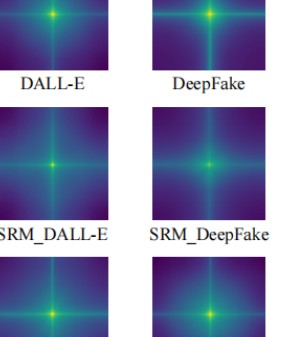
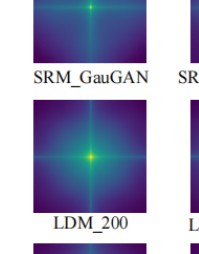

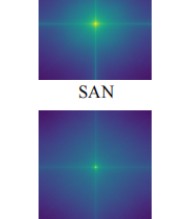
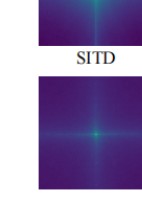

Figure 9: **Further spectrogram analysis.** We present additional spectrograms corresponding to more data. The filtering process attenuates the characteristics in the low-frequency regions, thereby enhancing the model's ability to learn and detect high-frequency features.

## C    T-SNE VISUALIZATION ANALYSIS OF HYPERDET AND UNIVFD

In Figure 10, we present the visualization comparison between our method, HyperDet, and the UnivFD method. Experiments were conducted on data from GAN models, diffusion models, and

autoregressive models. The results demonstrate that our method can almost perfectly distinguish the data from all models, highlighting the superior generalization capability of our approach.

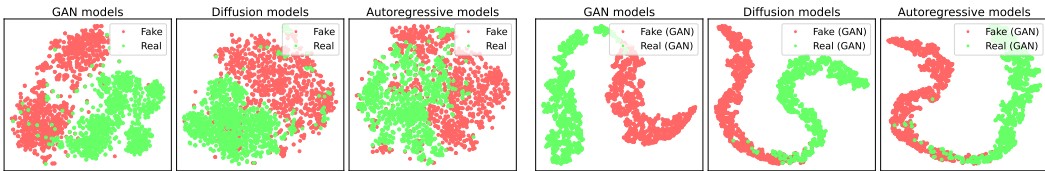

(a) t-SNE visualization of UnivFD

(b) t-SNE visualization of HyperDet

Figure 10: **t-SNE visualization analysis of HyperDet and UnivFD.** In Figure 10a, we present the t-SNE visualization results of the UnivFD method. It was observed that while the data from GAN models can be well distinguished, there are difficulties in differentiating data from diffusion and autoregressive models. Figure 10b shows the results of our method (HyperDet), which demonstrates excellent generalization ability across various generative models.

# D   FURTHER ADVANTAGES OF HYPER LORAS AND MODEL MERGING IN PERFORMANCE OPTIMIZATION

## D.1   COMPARISON OF THE PERFORMANCE BETWEEN HYPER LORAS AND THE COMBINATION OF MOE WITH LORA

Our approach employs a Hyper LoRAs to generate the corresponding number of LoRA modules for fine-tuning. However, we also considered the possibility of fine-tuning using a combination of MoE and LoRA. What kind of results could this fine-tuning strategy achieve? As shown in Table 11, the Hyper LoRAs method is capable of learning more shared knowledge, thereby achieving superior performance. Furthermore, the Hyper LoRAs can complete the entire training and inference process in a single step, unlike MoE, which requires sequential training. Data shows that LoRA generated by the Hyper LoRAs demonstrates superior performance, primarily because the hypernetwork effectively learns and captures the feature differences between various LoRA networks. As a result, it exhibits greater flexibility and accuracy in complex tasks. In contrast, while MoE LoRA fine-tuning also shows competitiveness in specific tasks, its performance largely depends on the structure of the expert models and the complexity of the task.

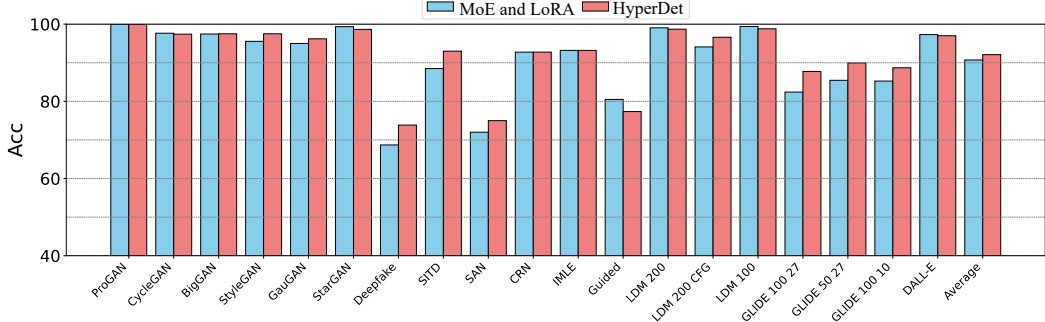

Figure 11: **Comparison of Accuracy Between Hypernetwork-Generated LoRA Fine-Tuning and MoE LoRA Fine-Tuning.** This study conducts a comparative analysis of the accuracy achieved through two different LoRA fine-tuning methods: one generated by a Hypernetwork and the other using a Mixture of Experts (MoE) model. The Hypernetwork-based approach dynamically generates LoRA networks based on specific embedding parameters, providing adaptive fine-tuning capabilities. Experimental results show that the Hypernetwork structure yields an average accuracy improvement of +1.38%.

## D.2 COMPARISON OF THE PERFORMANCE BETWEEN MERGED AND NON-MERGED MODELS

Our previous experiments have demonstrated that model merging can achieve significant results. However, does each LoRA generated by the Hyper LoRAs already perform well on its own? This section presents experiments showing that model merging significantly enhances the overall model's generalization detection capability. Figures 12 and 13 demonstrate that our merging method achieves the best generalization performance on the UnivFD dataset. As shown in the density plots, the detection results after merging exhibit a high-density distribution in the region of high accuracy, further validating the effectiveness and stability of this method across different scenarios.

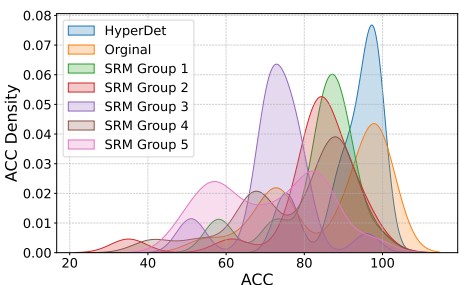 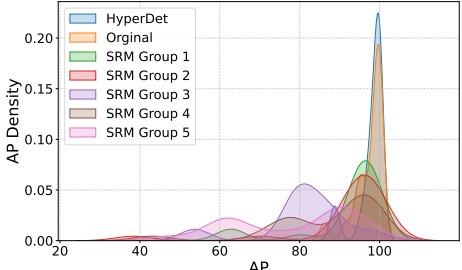

Figure 12: **Accuracy density distribution under different merging methods.** We present the accuracy distribution for the merging method, the original images (without filtering), and five separate filter groups. Under our merging method, there is a significant concentration in the region of high accuracy.

Figure 13: **Average precision density distribution under different merging methods.** We present the average precision distribution for the merging method, the original images (without filtering), and five separate filter groups. Under our merging method, the region with higher average precision exhibits a significant concentration.

## E AVERAGE PRECISION UNDER DIFFERENT NUMBERS OF LAYERS AND RANK

Table 4: **Average precision under different numbers of layers and ranks.** As shown in Table 4, average precision also demonstrates a high level of stability.

| Fine-tuned layers | LoRA rank | Pro-GAN | Cycle-GAN | Big-GAN | Style-GAN | Gau-GAN | Star-GAN | Deep-fakes | SITD | SAN | CRN | IMLE | Guided | 200s | 200s w/CFG | 100s | 100-27 | 50-27 | 100-10 | DALL-E | mAP |
|---|---|---|---|---|---|---|---|---|---|---|---|---|---|---|---|---|---|---|---|---|---|
| | 4 | **100.0** | 99.95 | 99.93 | 99.48 | 99.89 | **100.0** | 86.03 | 98.53 | 86.76 | 99.24 | 99.91 | 96.59 | 99.88 | 99.24 | 99.87 | 96.46 | 97.67 | 97.27 | 99.60 | 97.70 |
| 7 | 8 | **100.0** | 99.92 | 99.95 | 99.81 | 99.94 | 99.99 | 92.61 | 97.42 | 88.55 | 99.42 | 99.96 | 95.11 | 99.90 | 98.85 | 99.89 | 95.83 | 97.38 | 96.71 | 99.64 | **97.94** |
| | 16 | **100.0** | 99.97 | 99.92 | 99.78 | 99.89 | **100.0** | 86.01 | 92.22 | 87.83 | **99.43** | 99.99 | 97.05 | 99.94 | 99.19 | 99.94 | 96.86 | 97.99 | 97.73 | 99.66 | 97.55 |
| | 32 | **100.0** | 99.94 | 99.91 | 99.66 | 99.92 | 99.99 | 87.48 | 96.20 | 88.87 | 99.22 | 99.96 | 95.63 | **99.95** | 99.13 | 99.91 | 96.84 | 97.84 | 97.68 | 99.63 | 97.78 |
| | 4 | **100.0** | **99.98** | 99.90 | 99.84 | 99.92 | **100.0** | 89.57 | 97.91 | 83.61 | 97.17 | 99.96 | 94.68 | 99.77 | 98.95 | 99.84 | 93.33 | 95.63 | 94.89 | 99.60 | 97.08 |
| 8 | 8 | **100.0** | **99.98** | 99.90 | 99.88 | 99.95 | **100.0** | 92.65 | **98.97** | 84.84 | 97.99 | 99.97 | 93.97 | 99.77 | 98.77 | 99.84 | 93.17 | 95.35 | 95.05 | 99.40 | 97.34 |
| | 16 | **100.0** | 99.96 | 99.89 | 99.73 | 99.93 | **100.0** | 88.38 | 97.12 | 89.22 | 98.82 | 99.98 | 95.31 | 99.86 | 99.14 | 99.90 | 97.20 | 97.99 | 98.02 | 99.65 | 97.90 |
| | 32 | **100.0** | **99.98** | 99.93 | 99.86 | 99.95 | **100.0** | 92.05 | 98.33 | 79.84 | 98.19 | 99.98 | 92.32 | 99.70 | 98.67 | 99.75 | 90.94 | 94.09 | 92.95 | 99.14 | 96.61 |
| | 4 | **100.0** | 99.96 | 99.92 | 99.26 | **100.0** | 99.99 | 80.57 | 95.32 | 90.93 | 98.43 | 99.97 | 94.04 | 99.81 | 99.15 | 99.88 | 91.31 | 94.48 | 93.71 | 99.42 | 96.64 |
| 9 | 8 | **100.0** | 99.96 | 99.93 | 99.70 | 99.97 | **100.0** | 86.07 | 96.43 | 85.40 | 96.51 | **100.0** | 94.49 | 99.86 | 99.12 | 99.91 | 91.80 | 95.31 | 94.10 | 99.42 | 96.74 |
| | 16 | **100.0** | **99.98** | **99.98** | **99.90** | 100.0 | **100.0** | 89.77 | 93.22 | **91.39** | 95.28 | 99.90 | **97.26** | 99.94 | **99.75** | **99.98** | **97.41** | **98.41** | **98.34** | **99.83** | 97.91 |
| | 32 | **100.0** | 99.97 | 99.96 | **99.90** | 99.96 | **100.0** | 86.51 | 97.89 | 88.91 | 92.42 | 99.98 | 96.30 | 99.89 | 99.52 | 99.94 | 93.99 | 96.69 | 95.77 | 99.76 | 97.22 |

## F EXPOSITION OF THE ALGORITHM

---

**Algorithm 1** HyperDet Training

---

**Input:** An image data $x$

Apply SRM filters group to obtain 5 different image variations $x_1, x_2, x_3, x_4, x_5$, and the original image $x_6$

**for** each image data $x_i$ where $i = 1, 2, \ldots, 5$ and the original image $x$ **do**

    Feed $x_i, x$ into the model

    Use the corresponding Hyper LoRAs $h(t_i, L_j, P_k)$ to generate LoRA fine-tuning parameters

    Calculate the loss $L_i$ for each input $x_i$ as:

$$\mathcal{L}_{\text{bce}}^{F_i} = \mathcal{L}\left(f(x_i, \theta + \Delta\theta_i), y\right)$$

$$\mathcal{L}_{\text{bce}}^{O} = \mathcal{L}\left(f(x_6, \theta + \Delta\theta_6), y\right)$$

where $f(x_i, \theta + \Delta\theta_i)$ is the model output for input $x_i$ with LoRA fine-tuning parameters $\Delta\theta_i$, and $y$ is the ground truth.

    **Output:** Total loss $\mathcal{L}_{\text{total}}^{i} = \alpha\mathcal{L}_{\text{bce}}^{O} + (1 - \alpha)\mathcal{L}_{\text{bce}}^{F_i}$

    Perform backpropagation using the total loss $\mathcal{L}_{\text{total}}^{i}$ to update the model parameters

**end for**

---

**Algorithm 2** HyperDet Detection

---

**Input:** An image data $x$

Apply SRM filters group to obtain 5 different image variations $x_1, x_2, x_3, x_4, x_5$, and the original image $x_6$

Initialize output $y = 0$

**for** each image data $x_i$ where $i = 1, 2, \ldots, 5$ **do**

    **if** $i = 1$ **then**

        Feed both $x_1$ and $x_6$ into the model

        Use the corresponding Hyper LoRAs $h(l_1, L_j, P_k)$ to generate LoRA fine-tuning parameters

        Calculate the model output $y_1$ for $x_1$ and $x_6$ as:

$$y_1 = f(x_1, \theta + \Delta\theta_1) + f(x_6, \theta + \Delta\theta_6)$$

        Update the output: $y = y + y_1$

    **else**

        Feed only $x_i$ into the model

        Use the corresponding Hyper LoRAs $h(l_i, L_j, P_k)$ to generate LoRA fine-tuning parameters

        Calculate the model output $y_i$ for $x_i$ as:

$$y_i = f(x_i, \theta + \Delta\theta_i)$$

        Update the output: $y = y + y_i$

    **end if**

    **if** $y \geq$ threshold **then**

        Continue to the next $x_i$

    **else**

        Break the loop

    **end if**

**end for**

**Output:** Final output $y$

---

