# OpenReview forum: "HyperDet: Generalizable Detection of Synthesized Images by Generating and Merging A Mixture of Hyper LoRAs"
_ICLR.cc/2025/Conference — ICLR 2025 Conference Withdrawn Submission_

### Official Review · Reviewer_AX8T · 2024-11-03

**Soundness:** 3
**Presentation:** 3
**Contribution:** 3
**Rating:** 5
**Confidence:** 4

**Summary:**

In this paper, the authors present HyperDet, a novel and generalizable detection framework designed to effectively identify synthetic images. By integrating shared knowledge from lightweight expert detectors and leveraging a large pretrained vision model, HyperDet captures both general detection features and task-specific characteristics. The experiments on the UnivFD and Fake2M datasets show the effectiveness of their approach.

**Strengths:**

1. The authors present a novel approach to synthetic image detection with the introduction of HyperDet. Through the incorporation of hypernetworks that generate optimized weights for specialized LoRA experts, this approach enhances the extraction of generalized discernible artifacts.
2. The authors propose an SRM filter grouping strategy designed to capture varying levels of pixel artifacts based on functionality and complexity. They also introduce a novel objective function to balance pixel and semantic artifacts.
3. Experimental results on the UnivFD and Fake2M datasets demonstrate the framework’s effectiveness.
4. The paper is well-organized, presenting a coherent structure that enhances readability. The authors also provide detailed code, which supports reproducibility and further exploration of their work.

**Weaknesses:**

1. Lack of comparison experiments: The author selected different experimental comparison methods for different datasets UnivFD and Fake2M. The authors need to compare the same baseline method on different datasets to verify the effectiveness of the method.
2. Insufficient experimental analysis: HyperDet performed worse than some baseline methods on Fake2M datasets, but this was not explained at all in the experimental analysis.
3. The author claims to propose a novel objective function to balance the pixel and semantic artifacts effectively. However, this function is simply a weighted sum of the binary cross-entropy loss of the original image and the filtered image.

**Questions:**

1. My primary question is why the authors did not compare the baseline method UnivFD on the Fake2M dataset, as shown in Figure 4. It is clear that the UnivFD is the best performing baseline method on the UnivFD dataset, as shown in Table 2 and Figure 5.

2. In Figure 4, what do the baseline methods CLIT and CLIP_MoE correspond to in Tables 1 and 2, respectively? The Fake2M dataset obviously includes images generated by various diffusion models, making detection more challenging. However, the baseline methods used for the two datasets are not exactly the same, which makes it difficult for readers to be fully convinced of HyperDet's performance.

3. Why were five SRM filter groups selected? This part lacks an ablation study.

4. In Figure 5(b), why does HyperDet maintain the same or even increase the detection accuracy after the JPEG compression ratio of 90, while all other baseline methods continue to decline when detecting images generated by diffusion models?

---

### Official Review · Reviewer_XQiP · 2024-11-04

**Soundness:** 3
**Presentation:** 2
**Contribution:** 3
**Rating:** 5
**Confidence:** 2

**Summary:**

This paper presents a new AI generated image detection method, based on SRM filtering, which is evaluated on UnivFD and Fake2M datasets. Compared with several previous methods, the proposed method shows the best results on those two datasets as well as under image processing like Gaussian blurring. Overall, even though this paper shows more favorable results than others, the proposed method section is hard to follow and I am not sure other researchers can easily reproduce the experimental results reported in this paper. Besides, I also have some concerns regarding the experiments, as the proposed method leverages a more powerful network backbone, i.e., CLIP pretrained ViT model, which is stronger than previous applied backbone CNN in CNNSpot.

**Strengths:**

+ The proposed method shows stronger results on two recent public datasets. Besides, the proposed method also shows better results on different types of generative models, such as GAN, diffusion models, DALL-E, etc.

+ This paper leverages LoRA in AI generated image classification, which sounds interesting.

**Weaknesses:**

- Figure 1 is very confusing, especially the right part. It is very hard to understand the structure of the proposed model.

- From Figure 3, it seems that the proposed method relies on high frequency difference between real and fake images to recognize AI generated images. However, artifacts related to high frequency can be reduced by post-processing or regularization. Besides, many fake patterns are related to high level bias to real images, while the proposed method does not use.

- Leveraging LoRA is interesting, but this paper needs to discuss more about the motivation about using LoRA. For example, this paper needs to discuss more motivation and comparison to w/o using LoRA.

- This paper leverages CLIP pretrained ViT, while other methods utilize less powerful backbone. Is this the reason that the proposed method is more favorable results? I am not totally convinced by the experiments.

**Questions:**

- Why do you group SRM filters into 5 groups, instead of other numbers?

- The entire method section discuss a model with SRM filtering, including Figure 1, however, Eq 9 shows the proposed method is trained with original image as well. How does this method predict? Do you use raw images and filtered images together to detect fake images?

---

### Official Review · Reviewer_raxs · 2024-11-04

**Soundness:** 2
**Presentation:** 2
**Contribution:** 2
**Rating:** 3
**Confidence:** 4

**Summary:**

1.	This paper introduces a framework termed HyperDet for AI-Generated image detection.
2.	Extensive experiments demonstrate that HyperDet achieves state-of-the-art results on UnivFD and Fake2M benchmarks.

**Strengths:**

The method is effective in UnivFD and Fake2M benchmarks.

**Weaknesses:**

1.	Limited novelty: This method primarily involves a straightforward integration of high-level RGB information with low-level data, as outlined in [1] for AI-generated image detection. Additionally, the incorporation of high-level information is detailed in [2], while low-level information is explored in [3].
2.	The concept of utilizing multiple LoRAs is quite prevalent, as highlighted in [4], where their application in AI-generated image detection is discussed.
3.	Lack of inference throughput. What's the inference throughput of the proposed method? How is it compared with previous methods?
4.	Lack of extensive experiment results.There is a significant deficiency in experimental studies and results for key benchmarks, including GenImage, CNNDetection,DiffusionForensis, and AIGCDetectBenchmark.
[1]. A Sanity Check for AI-generated Image Detection
[2]. Towards Universal Fake Image Detectors that Generalize Across Generative Models
[3]. PatchCraft: Exploring Texture Patch for Efficient AI-generated Image Detection
4]. MoE-FFD: Mixture of Experts for Generalized and Parameter-Efficient Face Forgery Detection

**Questions:**

See weakness

---

### Official Review · Reviewer_LpT4 · 2024-11-04

**Soundness:** 2
**Presentation:** 3
**Contribution:** 2
**Rating:** 5
**Confidence:** 4

**Summary:**

In summary, the paper presents HyperDet, a novel and effective framework for detecting synthesized images with high generalization capabilities. By leveraging grouped SRM filters, Hyper LoRAs tuning, and model merging, HyperDet achieves competitive performance on multiple benchmarks.

**Strengths:**

The paper is written with excellent clarity. The motivation, approach, and experiments are well-explained. The figures and tables effectively complement the text and aid in understanding the methodology and results. The organization and flow of the paper are logical, making it easy to follow for readers.

**Weaknesses:**

1.The paper criticizes NPR for primarily relying on low-level pixel artifact features and neglecting semantic information, leading to high false positive rates. However, this criticism may be misplaced. NPR is designed based on the upsampling operations used in synthetic image generation, which are universal and generalizable across different models.The paper fails to provide sufficient evidence or analysis to support its claim that NPR's method leads to high false positive rates due to neglecting semantic information.
2. The proposed method combines low-level and semantic features (A+B) and uses a mixture of experts (MoE) for selection. While this approach aims to address the limitations of methods that rely solely on either low-level or semantic features, the paper lacks sufficient motivation and insight into why this combination is novel and significant. Simply combining existing techniques without providing a compelling rationale or analysis of the advantages over prior work may not meet the bar for novelty and significance required by ICLR.
3. The results achieved by the authors for NPR differ significantly from those reported in the original NPR paper, particularly in the context of Diffusion Models, where NPR reportedly attains an average accuracy of 95.2. We suspect that the use of a 20-class ProGAN dataset for training may have contributed to this discrepancy. While the authors have retrained a version of NPR using this dataset, it appears that this retrained model does not perform as well as the officially released NPR checkpoint.
4. The proposed method has a significantly higher parameter and computation cost compared to other methods, such as UnivFD and NPR. With a reasoning computation cost at least 6 times that of UnivFD and a much larger parameter count than NPR's 1.44M parameters, the method may be impractical for real-world applications.
5. Limited Test Set Coverage:The test set used in the evaluation has some limitations. Notably, it does not include GenImage or the latest diffusion model architectures (Flux, SD3, PixArt). The method's performance against these types of synthetic images remains unknown. Expanding the test set to include these models would provide a more comprehensive evaluation and demonstrate the generalizability of the approach.
6. Missing Baselines: The paper omits comparisons with specific baselines referenced in [1-3].

[1] Forgery-aware Adaptive Transformer for Generalizable Synthetic  Image Detection
[2] FakeInversion: Learning to Detect Images from Unseen  Text-to-Image Models by Inverting Stable Diffusion
[3] Improving Synthetic Image Detection Towards Generalization: An Image Transformation Perspective

**Questions:**

Please refer to weakness

---

### Official Review · Reviewer_gNqm · 2024-11-04

**Soundness:** 1
**Presentation:** 3
**Contribution:** 2
**Rating:** 1
**Confidence:** 4

**Summary:**

The paper proposes HyperDet, a mixture of experts approach towards the detection of generated images. The basic idea of the introduced method is to apply a pre-processing with groups of Spatial Rich Model (SRM) Filters which are used as initial feature extractors before group specific LoRA fine-tunings of pre-trained ViT hyper-backbones. In a final step, the the results of the different groups are then merged to obtain the prediction.
The paper claims SOTA results on two recent detection benchmarks (UnivFD and Fake2M) with improved generalization abilities toward unseen generators.

**Strengths:**

Iff (if and only if) the presented results hold in a popper evaluation (see weaknesses), the proposed method would lead to a significant improvement of the current (unfortunately heavily biased (see weaknesses) SOTA results on two recent benchmarks.

**Weaknesses:**

Unfortunately, the paper suffers from severer issues both at a conceptual level and major technical problems in the evaluation part.

The problems begin with the motivation: the paper motivates the need for the detection of generated imaged images simply by stating that " the proliferation of such [generated] images poses serious threats to public opinion and the authenticity of information", without any scientific evidence of this central claim. While there indeed might be good reasons to motivate this work, they should be backed by some evidence (citations!). Further, the paper does not make the slightest attempt to reflect if the classification of "fake" images (information) could be harmful in any way -> see ethics concerns.

The central argument against this paper (in it's current state) is the shockingly naive definition (or better the lack of a definition) of what "real" images are actually supposed to be. Following an unfortunate long line of recent publications, this paper not only lacks such a definition, but also fails to take modern image capturing pipelines and their biases into account. While the question of "what is a real image" might sound quite philosophical, there concrete technical aspects to consider: modern cameras in smartphones and surveillance devices (which make up the majority of "photo taken" today), are NOT simple optical projections of the real (3D) world onto a 2D plane anymore. Instead, they take multiple projections through multiple lenses and over different time segments which are then computed into a "photo" (see [1] for an overview). This process involves the application of neural networks which are quite similar in they architecture to the generative networks used to produce "fake" images. Hence, a clear definition should state if "real" means "not processed by a NN" - and if this the case, modern smart phone images must be included in the "real" or "fake" test set to guarantee a clean validation.

Beyond the question of "what is real", the "real" training  data must also be selected with utmost care in order to avoid additional biases [2] like compression, image size and image content, which easily render any evaluation of detector algorithms worthless - [2] pointed out that most generation detectors are actually jpg and/or size detectors.
Unfortunately, the paper does not provide ANY information about the "real" data used for training and evaluation (beyond the number of images used) -> see questions.

Even worse, the paper does not provide any results (precision / recall) for the "real" class. Hence, the reported improvements in generalization could simply be the result of a very low recall of "real" images. Since "real" images are still (depending on the definition) much more frequent than generated images, poor performance on this calls would make the entire approach invalid.

[1] Delbracio, Mauricio, et al. "Mobile computational photography: A tour." Annual review of vision science 7.1 (2021): 571-604.
[2] Grommelt, Patrick, et al. "Fake or JPEG? Revealing Common Biases in Generated Image Detection Datasets.", ECCV 24

**Questions:**

* give a solid definition of "real" -> does this include modern image pipelines? Justify your "real" data curation for training and test
* describe the "real" data in test and training in detail. What are the sources? How did you sample the data? What are the data properties regarding known biases (jpg, size, content)?
* report precision and recall for the real class in every experiment

**Details Of Ethics Concerns:**

Without a proper definition (if this at all possible) of "real", "fake" detection algorithms are potential tools of censorship. The ability to detect and ban generated content has the potential to harm free speech like regime critical memes. This should at least be reflected by the authors and discussed in the paper.

---

### Note · Authors · 2024-11-13

I have read and agree with the venue's withdrawal policy on behalf of myself and my co-authors.